# Using Multi-Resolution Satellite Data to Quantify Land Dynamics: Applications of PlanetScope Imagery for Cropland and Tree-Cover Loss Area Estimation

**Jeffrey Pickering \*, Alexandra Tyukavina** 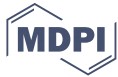**, Ahmad Khan, Peter Potapov, Bernard Adusei, Matthew C. Hansen** **and André Lima**

Department of Geographical Sciences, University of Maryland, College Park, MD 20742, USA; atyukav@umd.edu (A.T.); akhan234@umd.edu (A.K.); potapov@umd.edu (P.P.); badusei@umd.edu (B.A.); mhansen@umd.edu (M.C.H.); delima@umd.edu (A.L.)

\* Correspondence: jeffreyp@umd.edu

**Abstract:** The Planet constellation of satellites represents a significant advance in the availability of high cadence, high spatial resolution imagery. When coupled with a targeted sampling strategy, these advances enhance land-cover and land-use monitoring capabilities. Here we present example regional and national-scale area-estimation methods as a demonstration of the integrated and efficient use of mapping and sampling using public medium-resolution (Landsat) and commercial high resolution (PlanetScope) imagery. Our proposed method is agnostic to the geographic region and type of land cover and change, which is demonstrated by applying the method across two very different geographies and thematic classes. Wheat extent is estimated in Punjab, Pakistan, for the 2018/2019 growing season, and tree-cover loss area is estimated over Peru for 2017 and 2018. We used a time series of PlanetScope imagery to classify a sample of $5 \times 5$ km blocks for each region and produce area estimates of 55,947 km$^2$ ($\pm 9.0\%$) of wheat in Punjab and 5398 km$^2$ ($\pm 9.1\%$) of tree-cover loss in Peru. We also demonstrate the use of regression estimation utilizing population information from Landsat-based maps to reduce standard errors of the sample-based estimates. Resulting regression estimates have SEs of 3.6% and 5.1% for Pakistan and Peru, respectively. The combination of daily global coverage and high spatial resolution of Planet imagery improves our ability to monitor crop phenology and capture ephemeral tree-cover loss and degradation dynamics, while Landsat-based maps provide wall-to-wall information to target the sample and increase precision of the estimates through the use of regression estimation.

**Keywords:** PlanetScope imagery; Pakistan; wheat; Peru; tree-cover loss; degradation; land-cover; sampling

## 1. Introduction

Mapping changes in land cover and land use is one of the most important and highly researched topics in remote sensing [1]. The tools of remote sensing science are an integral part of implementing national and sub-national scale mapping programs in accordance with the guidelines laid out by the Intergovernmental Panel on Climate Change (IPCC) [2]. Fully automating this mapping process is difficult because of a number of factors, including the diversity of land-cover change types [1]. Despite this diversity, national scale mapping requires repeatable and consistent change area estimates over different land covers [2]. By coupling statistical sampling with classifying high cadence, circa 4m satellite imagery we present a generic method for estimating land cover and change area, demonstrating its utility in two very different geographies and target classes: tree-cover loss in Peru and wheat area in Pakistan.

### 1.1. Sampling Background

Advances in remote sensing technology provide increasingly better opportunities to map and track changes in land cover and land use. One of the most well recognized and

efficient ways to monitor these changes is by using satellite imagery to map land-cover change by employing a probability sample of the target area. Good guidance on area estimation from the IPCC recommends methods such as probability sampling that neither under- nor overestimate and that reduce uncertainties as far as practicable [2].

There is a rich history of publications on probability sampling approaches to estimating area for land-cover and land-use applications [3–9]. A major benefit to sampling is the significant time and cost savings as opposed to mapping an area wall-to-wall, as only a portion of the target region needs to be mapped [3,5]. There is no universally ideal sampling design because of the many decisions relating to sample block size, mapping protocol, and reference imagery, which depend on cost/effort limitations, local geographical considerations, and specific study objectives [3]. At the national and continental scales, sample designs that employ 'stratified random sampling' often represent the most efficient trade-off between achieving a desirable accuracy and meeting design criteria [3,9]. Stratified random sampling focuses the analysis on homogeneous areas with respect to the target variable. While many previous studies have important differences, a number of them employ the following steps:

- Attributing a value to a population of regular shaped blocks based on some auxiliary variable;
- Stratifying the population of blocks according to the attributed value into several strata and selecting a random sample of these blocks from each stratum;
- Mapping the land-cover change of the target variable over a defined time period using historical satellite imagery;
- Producing statistical estimates and uncertainties of target variable area and/or area change over the entire population and conducting validation fieldwork where appropriate.

Studies that broadly employ this approach include crop area estimates for soybean and corn in the continental US [10,11], and for soybean in South America [12]. Other studies employ variations on this approach for tree-cover loss area estimates over a given time period in the tropical biomes [5–8,13,14].

### 1.2. Satellite Imagery

The choice of reference satellite imagery is a key decision in the implementation of mapping programs. Ideally, reference imagery should be of sufficient spatial and temporal resolution such that the maps and estimates produced are highly accurate and relevant at both the local and national scales. For the studies looking at the historic land-cover dynamics, Landsat imagery is the best source of reference imagery, with its archive going back to the 1980s. However, for some applications, Landsat's 30 m resolution and a revisit interval of 8–16 days is not sufficient. For example, the forest degradation associated with partial canopy cover removals from selective logging might not be adequately captured in 30 m resolution data. Considerable uncertainty around the definition of forest degradation still exists; however, degradation is essentially tree-cover loss that occurs below some minimum mapping units. The minimum mapping unit is largely a function of the image resolution. Correctly quantifying the area affected by selective logging is also a function of having a reference image close to the date of disturbance. This is particularly significant for regions with persistent cloud cover, such as the tropics, where a weekly or bi-weekly satellite revisit date could result in months without usable imagery. Satellites with a high revisit frequency have a better chance to image an area that may have very few relatively cloud free days.

PlanetScope imagery is a 'game changer' in terms of both spatial and temporal resolution [15]. PlanetScope is a constellation of over 200 satellites that provide daily global image coverage at a 3–5 m spatial resolution. The normal scene size is approximately 24 km × 7 km, but this can vary with altitude [16]. Imagery at this resolution and revisit frequency allows for the mapping of larger clear-cut areas in forests [15] and importantly

increases our ability to track small-scale ephemeral changes in tree cover, such as selective wood harvesting.

The potential applications of PlanetScope imagery are not limited to tree-cover monitoring. In the agricultural field, it may include the potential to estimate pasture biomass in crop-livestock systems [17]. PlanetScope imagery also increases our ability to capture the full phenology of crop growth and intra-field variation that may be unclear at coarser temporal and spatial resolutions. Accurately capturing these types of short-term changes is an increasingly important aspect of land change mapping initiatives [15].

### 1.3. Study Objectives

Our study covers two distinct biomes and geographies. Firstly, we mapped winter wheat over a stratified random sample of blocks to estimate wheat area grown over the five-month-long growing season from November/December 2018 to March/April 2019 in Punjab, Pakistan [18]. Additionally, we conducted fieldwork in Punjab to compare the field-based estimates with the PlanetScope based estimates.

Secondly, we conducted tree-cover loss mapping over a stratified random sample of the humid tropical forest biome of Peru for the two calendar years of 2017 and 2018. These dates were chosen to align with the goals of national scale mapping programs, which typically aim to map tree-cover loss over a calendar year. The objective of this mapping was to assess the suitability of PlanetScope imagery to map tree-cover loss over a sample area to derive national scale tree-cover loss estimates.

We aimed to produce area estimates and uncertainties for both regions in accordance with IPCC good practice guidelines. This was done primarily to assess the utility of the PlanetScope imagery by mapping two different phenomena over two different land covers occurring in two different biomes. Efficient and robust methods to derive area estimates at scale based on the mapping of 3–5 m resolution satellite imagery are relevant to policy makers and land managers globally.

We also aimed to demonstrate the utility of wall-to-wall medium resolution maps (derived from Landsat) to increase precision of the sample-based estimates via regression estimation. Regression estimation utilizes existing correlation between target (high resolution estimates from PlanetScope and field data) and auxiliary (Landsat-based maps) variables for the sampled blocks, and population information (for all blocks) from auxiliary variable, which results in the increased precision of area estimates.

## 2. Materials and Methods

### 2.1. Description

Punjab province lies in northeastern Pakistan, covering the desert and temperate grasslands of the Indus river valley. It has a continental climate and can experience significant variations in temperature within short time periods. The fertile land of Punjab is particularly well suited for agriculture. Punjab sits between 27°N, 71°E and 34°N, 73°E. Relative to Peru, the northern latitude of Punjab results in a higher frequency of cloud-free image observations.

The humid tropical forest (HTF) zone in Peru sits to the northeast of the Andean divide, which runs from Ecuador in the northwest to Bolivia in the southeast. The HTF zone extends south from the equator to −15°S and sits between −68°E and −79°E. Figure 1 shows the locations and broad level biomes of both study areas.

### 2.2. Sample Design

We used a variation of the sampling design based on previously published work [3–8]. In both study areas, the populations consist of a regular grid of 5 km × 5 km equal area blocks. We selected a 5km block size based on the following considerations: (1) large enough to capture areas with potential errors of omission surrounding the class of interest mapped with Landsat; (2) compact enough for mapping with high-resolution data and for field validation. For PlanetScope imagery, a 5 km × 5 km block size allows for a balance

between good landscape coverage and the practicalities of dense time-series mapping and the associated data volumes.

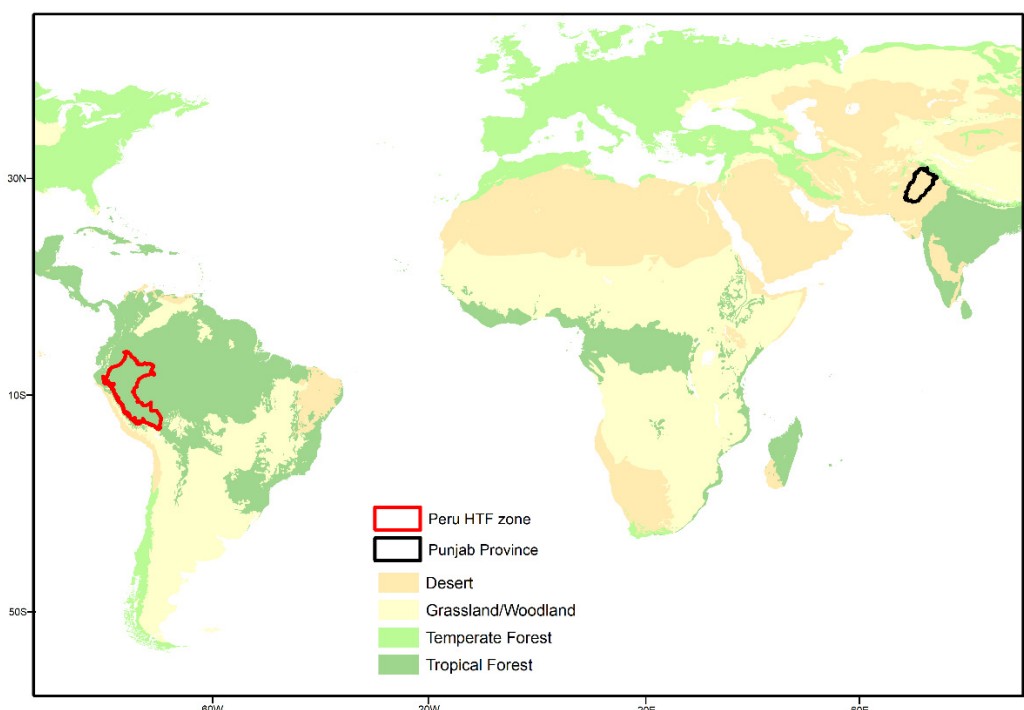

**Figure 1.** High Level terrestrial biomes and selected study areas in Pakistan and Peru.

### 2.2.1. Pakistan Sampling

For wheat area mapping in Punjab, only blocks that had ≥25% of the block within the province boundary were included. The auxiliary variable used was a previously produced wall-to-wall 2018 growing season map from Landsat [18,19]. By selecting blocks that contained over 200 hectares of wheat, we found that 67% of the block population in Punjab covered 98.8% of the total wheat area in the province. We therefore assigned this 67% as the 'wheat' stratum and the remaining 33% as the 'no-wheat' stratum (Tables 1 and 2). A sample of 25 blocks was selected in the wheat stratum for mapping. Figure 2 shows the distribution of sample blocks across Punjab.

**Table 1.** Sample design for Pakistan study. Percent loss per block is from 2018 auxiliary wheat map used to assign block values for strata boundary allocation.

| Stratum | % Wheat per Block | Area of Wheat (% from Total) | Total Blocks (Nh) | Sample Size (n) |
|---------|-------------------|------------------------------|-------------------|-----------------|
| Wheat | 8–96% | 98.8% | 5562 | 25 |
| No Wheat | 0–8% | 1.2% | 2736 | - |

**Table 2.** Sample design for the Peru study. Percent loss per block is from Hansen et al. (2013) global map and was used for strata boundary allocation.

| Stratum | % Loss per Block | Area of Loss (% from Total) | Total Blocks (Nh) | Sample Size (n) |
|---------|------------------|-----------------------------|-------------------|-----------------|
| No loss | 0 | 0 | 13,314 | 20 |
| Low loss | 0–2.1% | 26.9 | 14,767 | 30 |
| High loss | 2.1–31.2% | 73.1 | 3563 | 30 |

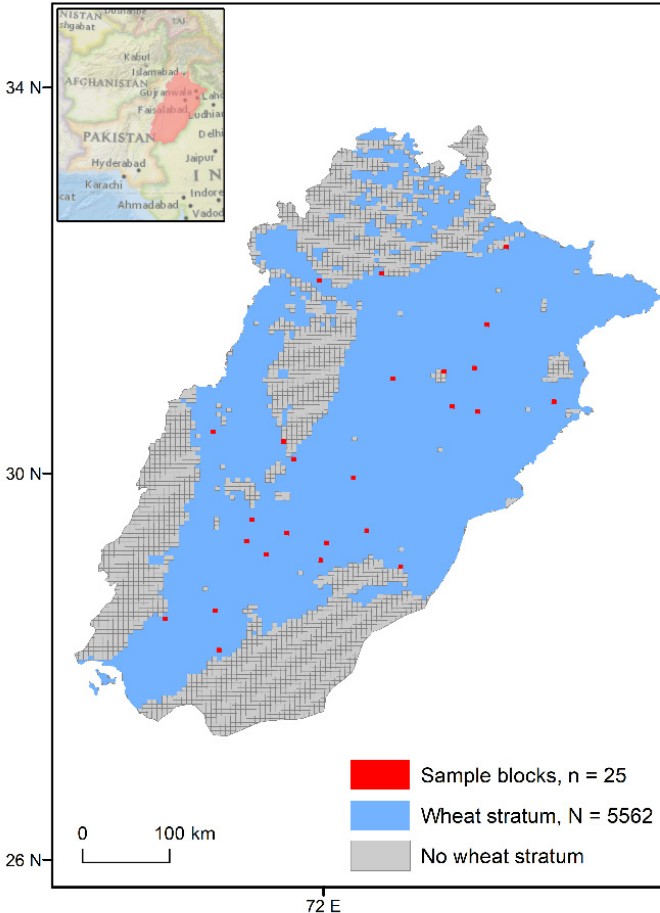

**Figure 2.** Distribution of selected sample blocks randomly selected within the wheat stratum across Punjab.

In addition to the mapping, validation fieldwork was conducted in Punjab over the 2018/2019 growing season. This allowed comparison of two independent area estimates, one derived from satellite imagery and one field based. The field visit included visiting 20 points in each of the 25 sampling blocks and recording land-cover information for each point.

### 2.2.2. Peru Sampling

For tree-cover loss mapping in Peru, only the blocks which had ≥40% of the block area within the humid tropical forest (HTF) biome of Peru were included. The boundary of HTF biome was received from the national government of Peru. Because 42% of the block population had no forest loss detected by the global forest loss map [20] in 2017 and 2018, these blocks were set aside into the separate "No-loss" stratum, and 20 sample blocks were selected from it (Figure 3). The rest of the blocks were subdivided into the "Low-loss" and "High-loss" strata based on per-block percent of 2017–2018 forest loss from the global map using Dalenius–Hodges rule, with resulting boundary between low and high-loss strata at 2.1% loss (Figure 3A). Sixty blocks were selected in the high- and low-loss strata following equal allocation of blocks between strata (30 blocks each, Figure 3B).

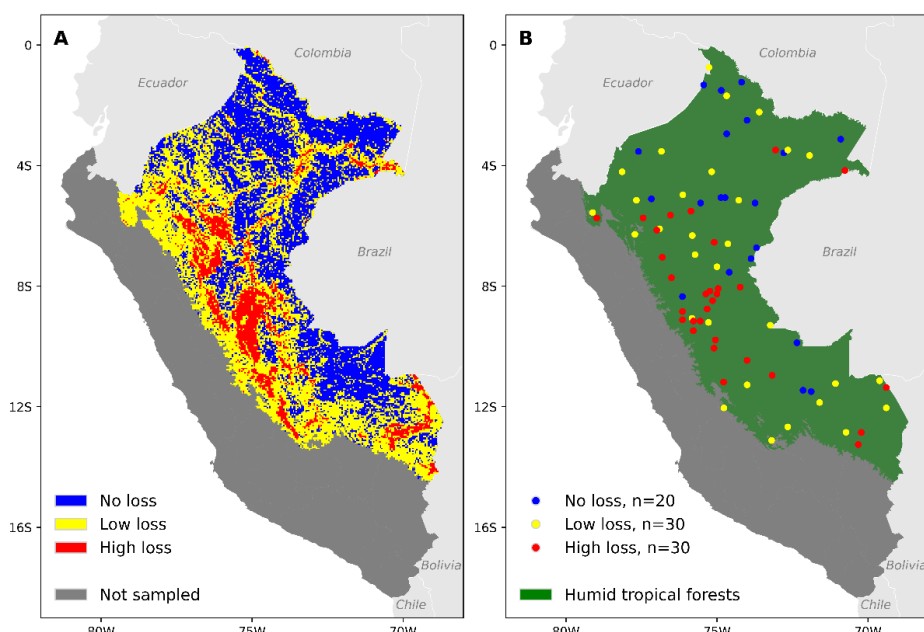

**Figure 3.** Sample design of Peru study: (**A**) sampling strata, (**B**) selected sample blocks. Strata boundaries in A correspond to the boundaries of 5 × 5 km equal area blocks; B shows centroids of selected 5 × 5 km blocks. Sampling was only performed within the humid tropical forest zone of Peru (green area in **B**).

*2.3. Reference Data*

PlanetScope daily imagery with circa 4 m per pixel spatial resolution was the primary source data for the block mapping in both studies. PlanetScope Analytic 4 band product was used, which includes three visual bands (red, green, and blue) and a near infrared band, and is orthorectified and calibrated to at-sensor radiance. PlanetScope data were made available through NASA's Commercial Smallsat Data Acquisition program, and selected using Planet's interactive image viewer and API (https://www.planet.com/explorer accessed on 1 January 2021) The Planet catalog provides the capability of uploading a region of interest and searching the catalog based on a target area. During a search of the Planet catalog for each sample block, we found near daily cloud free coverage of the Punjab region in Pakistan through the five month growing season of wheat. The growing season typically lasts from November/December to March/April [18,19]. Our imagery spans the dates 2 December 2018 to 29 April 2019.

In Peru, the imagery was much cloudier, but the daily revisit time meant we found approximately quarterly cloud-free coverage of every block. Notably, the radiometric and geometric quality of the PlanetScope imagery was noticeably better at the end of 2018 compared to earlier acquisition dates when images had radiometric and spatial alignment issues. For each block, we selected the last cloud-free image in 2016 and the first cloud-free image in 2019 to ensure we had the image closest to the start and end date of the study period. The downloaded imagery spans the period August 2016–April 2019. Images from the same day within each sample block were mosaicked together. There were 730 total scenes used to map the Peru blocks, whereas for Pakistan there were 2205 scenes. Less scenes were used for the Peru mapping because the data search was more selective. We used a higher threshold for cloud cover and aimed to get one cloud free image per quarter to ensure as even as possible distribution across the target time period. Figure 4 shows the number of scenes downloaded across the two-year period. The Pakistan imagery covered a shorter time period and there were more scenes downloaded in total with the intention being to cover the entire phenology of the wheat season.

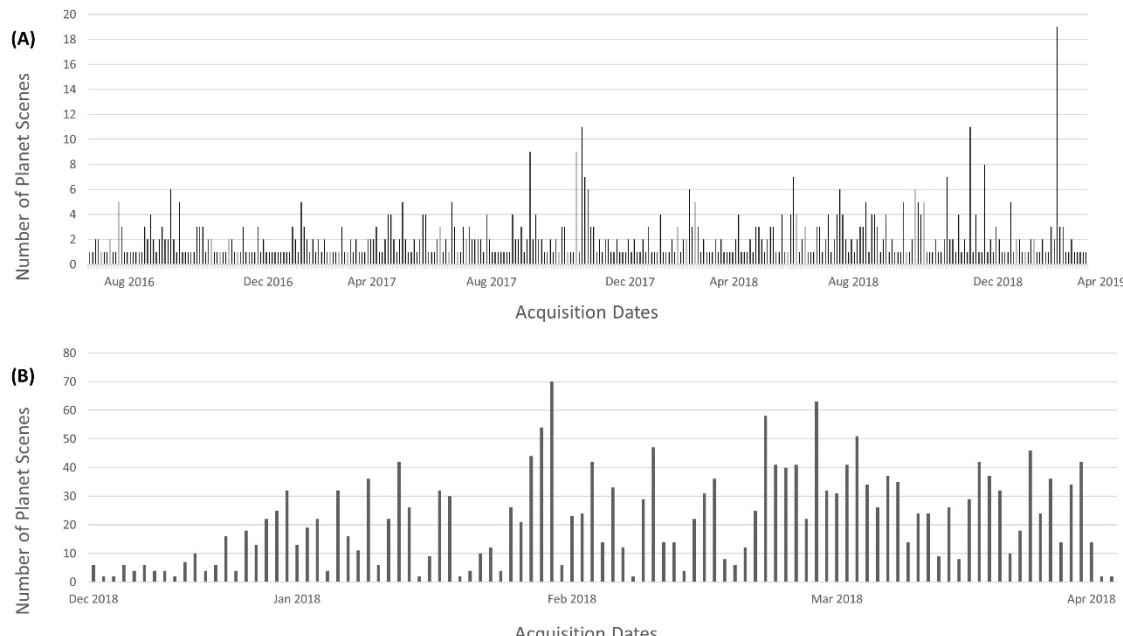

**Figure 4.** Acquisition timeline of the 730 planet scenes covering the two-year analysis period for Peru (**A**) and the 2205 planet scenes covering the four-month period over Punjab, Pakistan (**B**).

### 2.4. Block Mapping

We conducted a supervised classification of the wheat areas in Pakistan and of tree-cover loss areas in Peru using bagged classification trees [21]. The mapping involves cycling through the time series of imagery and drawing target and non-target training so the classification tree algorithm can delineate the boundaries between the two.

For wheat crop mapping in Punjab, this involved drawing target training over wheat crops and non-target training over all non-wheat land-cover types. In Peru, we draw the target training over easily identifiable tree-cover loss events and non-target training over adjacent land that had not experienced tree-cover loss. By employing this method, we are able to 'train-out' any cloud, atmospheric, or image misalignment issues manually in a user-driven fashion. Each block is mapped separately, so there is no need for standardized and normalized data inputs for all blocks. Figure 5 shows an example of the training over the PlanetScope reference imagery and the resultant maps after iteration.

Mapping of each sample block is an iterative process. Once the results of the first iteration were analyzed, the training data were modified as required until an accurate representation of the tree-cover loss or wheat area resulted. Training data were drawn from as many different scenes through the time period as possible to best capture the full extent of the wheat or tree-cover loss.

In Peru, we had a defined period for the two calendar years 2017 and 2018. However, the imagery covered the period late 2016 to early 2019. To ensure the loss we were mapping was as accurate as possible and only occurred within the target time period, we used Google Earth imagery and a small number of Sentinel 2 scenes where appropriate to narrow down the precise date of tree-cover loss events. Tree-cover loss occurring before or after our target period was trained as 'no loss' where possible; however, some amount of temporal ambiguity related to the timing of available anniversary date (start, end) imagery is still present in the final classification results. Because we are mapping temporally from multiple images, our final map products are not representative of any one image but are rather a combination of change in the target variable throughout the two-year period.

The image interpretation expert controlled the map quality for each sample block, and the map was considered final when all areas of the target class (wheat in Punjab and tree-cover loss in Peru) were correctly mapped. This way, the image classification results

are as good as the results of manual target class mapping using visual image interpretation. Machine learning classification improved the efficiency of the mapping without degrading the quality of the results. The relatively small size of the sample blocks simplified the classification and facilitated the high map quality.

For the Peru case study, in addition to mapping forest loss, we manually attributed forest loss areas to anthropogenic forest clearing and natural forest disturbances.

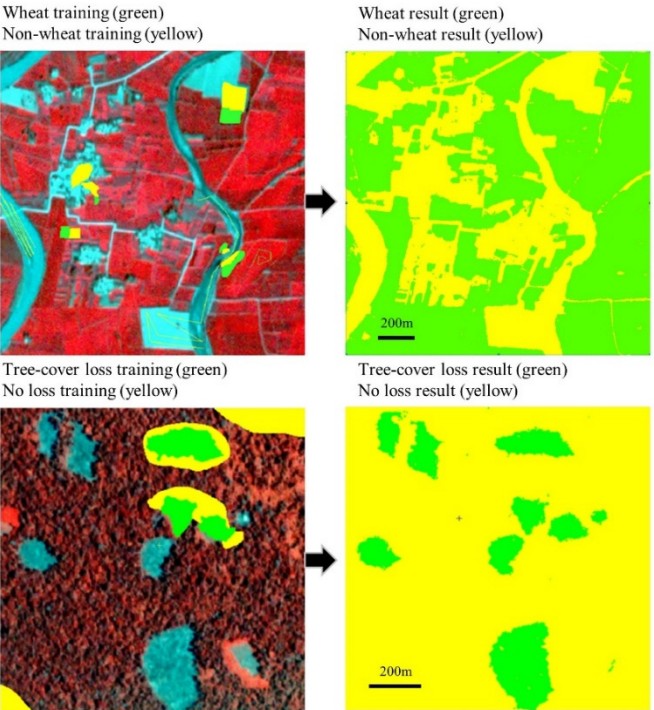

**Figure 5.** Examples of training drawn over PlanetScope imagery (4 m) and output from the iteration of bagged classification trees algorithm.

### 2.5. Area Estimates

After the mapping was complete, the proportion of the target class (wheat or tree-cover loss) from the total area was calculated for each block. For the field-based estimates of wheat area in Punjab, the per-block proportion of wheat was derived from 20 field points, randomly allocated within each block. A direct estimate of the mean proportion of the target class was then produced using the following equation, separately for Punjab and Peru (Cochran, Equation (5.1)) [22]:

$$\overline{y}_{dir} = \sum_{h=1}^{H} \frac{N_h}{N} \overline{y}_h \tag{1}$$

where $H$—number of sampling strata in each study region;

$N$—total number of blocks in each study region;

$N_h$—total number of blocks in stratum $h$;

$y_i$—proportion of target class from the reference mapping of each sampled block within stratum $h$;

$\overline{y}_h$—mean of $y_i$ within stratum $h$.

The total area of the target class in each region (Punjab and Peru separately) was then estimated as:

$$\overline{A}_{dir} = A_{tot} * \overline{y}_{dir} \tag{2}$$

where $A_{tot}$—total area of each region (in hectares).

Variance of direct estimate of the mean target class proportion in each region was estimated as (Cochran, Equation (5.6)) [22]:

$$\hat{V}(\bar{y}_{dir}) = \frac{1}{N^2} \sum_{h=1}^{H} N_h^2 \left(1 - \frac{n_h}{N_h}\right) \frac{s_{yh}^2}{n_h}$$

(3)

where $s_{yh}^2$ is the sample variance of $y_i$ in stratum $h$, computed as:

$$s_{yh}^2 = \frac{\sum_{i=1}^{n_h}(y_i - \bar{y}_h)^2}{n_h - 1} = \frac{\sum_{i=1}^{n_h}(y_i^2) - \frac{\left(\sum_{i=1}^{n_h} y_i\right)^2}{n_h}}{n_h - 1}$$

(4)

Standard error of the target class area in each study region was computed as:

$$SE(\overline{A}_{dir}) = A_{tot} * \sqrt{\hat{V}(\bar{y}_{dir})}$$

(5)

In addition to direct sample-based estimates of the target class area we have also produced a regression estimate, designed to increase the precision of the estimate by utilizing correlation between the target variable and the auxiliary variable, available for the entire population. We used the per-block proportion of forest loss from 2017–2018 forest loss maps for Peru [20]. For Pakistan, we used the 2018 auxiliary wheat map.

Within-stratum least squares estimate of slope of linear regression between the target and the auxiliary variable was estimated using the following equation (Cochran, Equation (7.56)) [22]:

$$b_h = \frac{\sum_{i=1}^{n_h}(y_i - \bar{y}_h)(x_i - \bar{x}_h)}{\sum_{i=1}^{n_h}(x_i - \bar{x}_h)^2}$$

(6)

where $x_i$—per-block proportion of auxiliary variable for each sampled block within stratum $h$; $\bar{x}_h$—mean of $x_i$ within stratum $h$.

Intercept of the linear regression for each stratum was estimated using the following equation:

$$a_h = \bar{y}_h - b_h \bar{x}_h$$

(7)

Regression estimate of the mean proportion of the target class in stratified random sampling, when a separate regression estimate is computed for each stratum mean (Cochran, Equation (7.49)) [22], was computed separately for each study region using the following equation:

$$\bar{y}_{reg} = \sum_{h=1}^{H} \frac{N_h}{N} \bar{y}_{regh}$$

(8)

where $\bar{y}_{regh}$—linear regression estimate of the stratum mean proportion of the target class (Cochran, Equation (7.48)) [22], computed as:

$$\bar{y}_{regh} = \bar{y}_h + b_h(\overline{X}_h - \bar{x}_h)$$

(9)

where $\overline{X}_h$—population mean of per-block proportions of auxiliary variable for stratum $h$ (computed from all blocks in each stratum).

Regression estimate of the target class area in each study region was estimated as:

$$\overline{A}_{reg} = A_{tot} * \bar{y}_{reg}$$

(10)

Variance of regression estimate of the mean proportion of target class in each study region was estimated as (Cochran, Equation (7.51)) [22]:

$$\hat{V}\left(\overline{y}_{reg}\right) = \frac{1}{N^2} \sum_{h=1}^{H} \frac{N_h^2\left(1 - \frac{n_h}{N_h}\right)}{n_h} \left(s_{yh}^2 + b_h^2 s_{xh}^2 - 2b_h s_{xyh}\right) \tag{11}$$

where $s_{yh}^2$ and $s_{xh}^2$ are the sample variances of $y_i$ and $x_i$ in stratum $h$ and $s_{xyh}$ is sample covariance between $x_{hi}$ and $y_{hi}$ in stratum $h$, computed as:

$$s_{xyh} = \frac{\sum_{i=1}^{n_h} x_i y_i - n_h \overline{x}_h \overline{y}_h}{n_h - 1} \tag{12}$$

Standard error of the target class area in each study region was estimated as:

$$SE\left(\overline{A}_{reg}\right) = A_{tot} * \sqrt{\hat{V}\left(\overline{y}_{reg}\right)} \tag{13}$$

### 3. Results

*3.1. Wheat Mapping in Punjab, Pakistan*

We were able to reliably identify and map wheat and produce estimates and associated uncertainties. The results for the wheat mapping in Punjab are presented in Table 3.

**Table 3.** Direct expansion and regression estimates for 2019 wheat area in Punjab, Pakistan.

| | Direct Estimate | | | Regression Estimate | | |
|---|---|---|---|---|---|---|
| | Area (km$^2$) | SE (km$^2$) | SE (%) | Area (km$^2$) | SE (km$^2$) | SE (%) |
| Field Data Based | 53,284 | 6322 | 11.9 | 54,289 | 4035 | 7.4 |
| Planet Data Based | 55,947 | 5019 | 9.0 | 56,884 | 2043 | 3.6 |

From the planet imagery mapping using the direct estimator, we estimate 55,947 km$^2$ ($\pm$9%) of planted wheat for the 2018/2019 growing season. The estimate increased to 56,884 km$^2$, but the uncertainty reduced from 9% to 3.6% when we applied the regression based estimate. For the field work based direct estimate, we estimate 53,284 km$^2$ ($\pm$11.9%) of planted wheat for the 2018/2019 growing season. Again, the estimate increased to 54,289 km$^2$ but the uncertainty reduced from 11.9% to 7.4% when we applied the regression based estimate. We should note that our stratification covered 98.8% of the wheat grown in Pakistan so the true ground estimate is likely to be around 1.2% higher than the estimate presented in Table 3, which translates to approximately 640–690 km$^2$ of planted wheat.

We found the PlanetScope imagery was highly suitable for mapping wheat area in Punjab, Pakistan, with field-based sample-based estimates closely matching the estimates derived from PlanetScope imagery (Table 3). The near daily image acquisitions allowed us to closely monitor the phenological progression from a bare soil spectral signature when the wheat is planted/young in November/December, through the shooting and heading stages in February to flowering in March and full maturity in April. Figure 6 shows an example result from one of the blocks with a portion of the full time series of planet imagery and the resulting map of wheat coverage over one of the blocks in eastern Punjab province. This is an intensive wheat area. The non-wheat areas are mostly either populated areas, rivers, or roads.

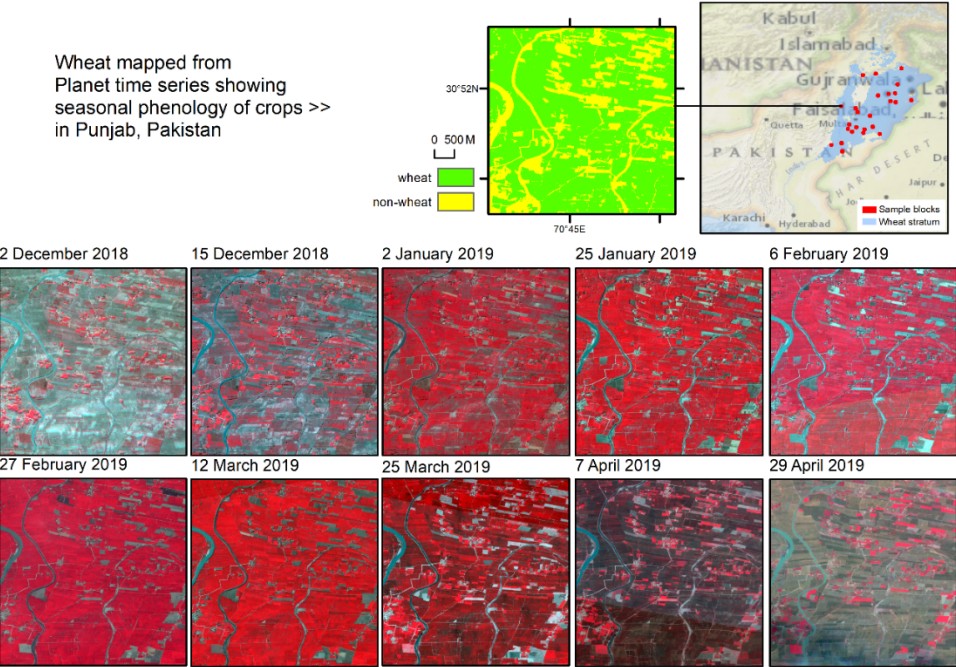

**Figure 6.** Planet time-series mapping over Eastern Punjab Province for one of the sample blocks.

*3.2. Tree-Cover Loss Mapping in Peru*

The results for tree-cover loss mapping in Peru are presented below in Table 4.

**Table 4.** Direct expansion and regression estimates for 2017–2018 tree-cover loss in Peru's humid tropical forests.

| | Direct Estimate | | | Regression Estimate | | |
|---|---|---|---|---|---|---|
| | Area (km$^2$) | SE (km$^2$) | SE (%) | Area (km$^2$) | SE (km$^2$) | SE (%) |
| All tree-cover loss | 5398 | 491 | 9.1 | 5121 | 263 | 5.1 |
| Anthropogenic loss | 4799 | 392 | 8.2 | - | - | - |

From the tree-cover loss mapping in Peru, our direct loss estimate was 5398 km$^2$ ($\pm$9.1%). This represents a tree-cover loss rate of 0.7% over the two year period within the Peruvian HTF zone. This is a rare land-cover dynamic, which would not have been adequately captured without a Landsat-based stratification. The average per-block loss in the high-loss stratum was 3.8%, and within the low-loss stratum it was 0.5%. No tree-cover loss was found in the no-loss stratum. Applying regression estimation resulted in a reduction of the standard error to 5.1%. The loss area estimate decreased from 5398 to 5121 km$^2$. Of the 5398 km$^2$ of tree-cover loss, 89% was anthropogenic (4799 km$^2$). Because the auxiliary variable does not distinguish anthropogenic and natural loss, we were not able to produce a regression estimate of anthropogenic loss alone.

Tree-cover loss events in Peru, such as selective logging, are fine scale, and can be temporary in nature. The dominant drivers of tree loss in our sample blocks are rotational shifting agriculture and selective tree harvest with associated infrastructure (roads, log landings, skid trails). Some larger-scale forest clearing for conversion from forest to industrial agricultural land uses was also present. Due to the time-series density, the PlanetScope imagery proved ideal for capturing both the larger-scale clearcuts and the ephemeral changes that disappear quickly. When mapping each block, the full time series of imagery covering the period 1 January 2017 to 31 December 2018 was used. Figure 7 shows a zoomed example of one of the mapped selective harvest sites.

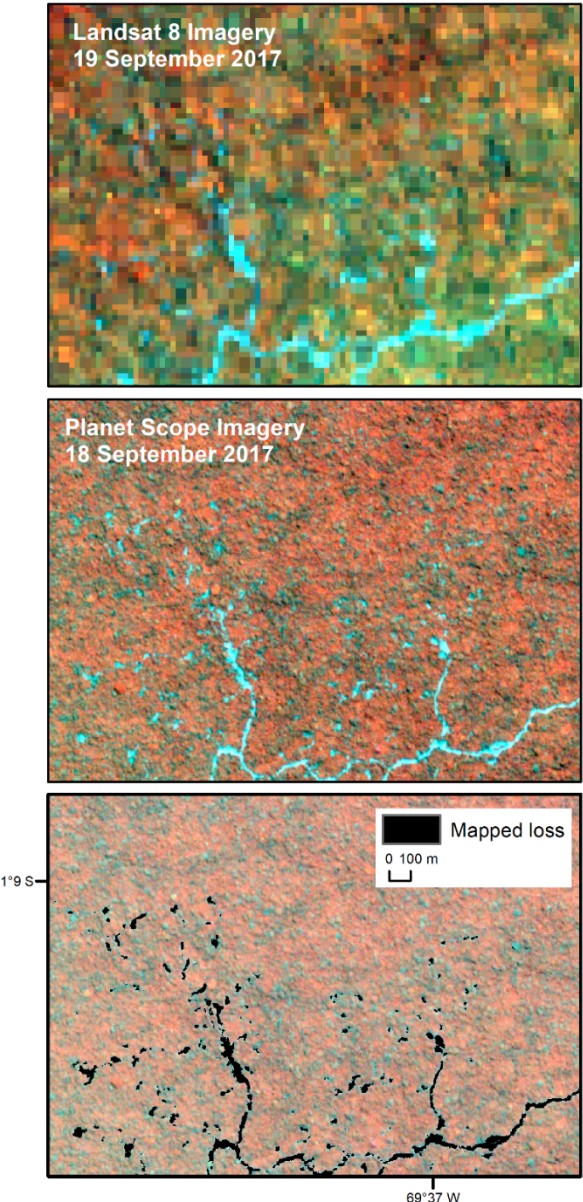

**Figure 7.** Mapped tree-cover loss comparison between Landsat and PlanetScope imagery acquired one day apart over a forestry operations site showing the canopy disturbances from selective logging and the associated roads and skid trails.

Figure 7 shows the nature of the fine scale detailed mapping that is afforded by the significant increase in resolution from Landsat to PlanetScope. We found that the resolution of PlanetScope imagery was sufficient to reliably capture tree-cover loss events as small as approximately 5 m across.

Figure 8 covers a riparian shifting agriculture area in the north of Peru. From the time series we can see the initial deforestation event and the regrowth occurring over the area within the study period, which would have led to an underestimation of tree-cover loss area if only the start and end of the period images were used for block mapping.

We found that, even in the relatively cloudy humid tropical forests of Peru, the time-series density of the PlanetScope imagery was sufficient to map ephemeral forest loss events.

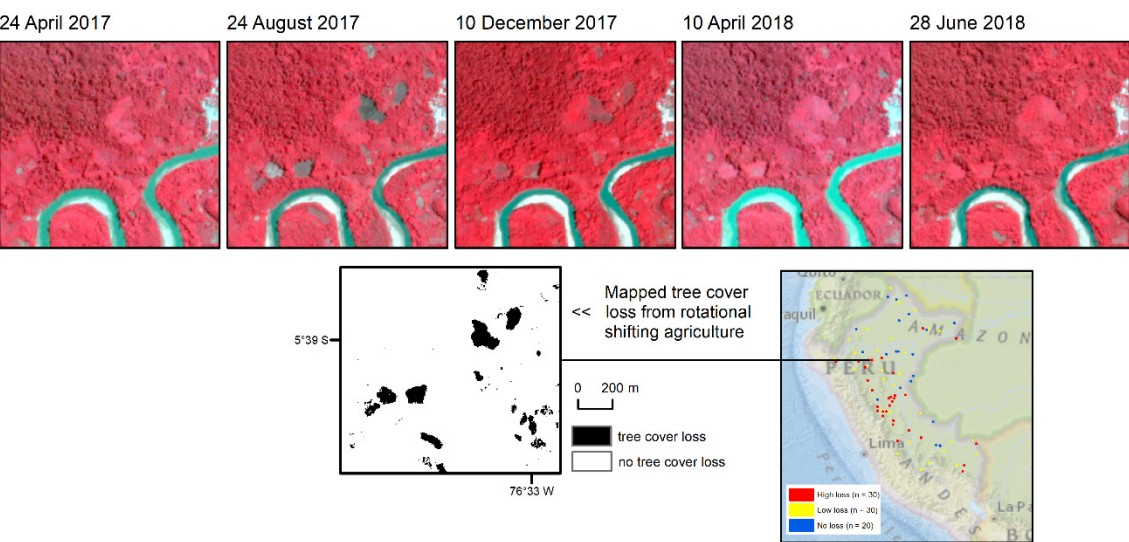

**Figure 8.** Planet time series over temporary tree-cover loss in riparian shifting cultivation area.

## 4. Discussion and Conclusions

We present an efficient use of multi-scale data in estimating the area of two important global land dynamics—commodity crop production and deforestation. The use of turnkey Landsat products, in the case of Pakistan, a customized annual wheat map extent algorithm for Punjab province, and for Peru, a subset of the Global Forest Watch annual forest loss product, provides an efficient means of targeting the theme of interest for subsequent stratified random sampling. The use of PlanetScope imagery as reference data enables improved characterization of wheat extent and tree-cover loss for the respective study areas. In the case of Pakistan, a robust phenological profile facilitates wheat characterization. In the case of Peru, the high cadence of Planet acquisitions ensures the inclusion of short-term disturbances such as selective logging. The resolution of PlanetScope data affords a more discrete mapping of the small-scale spatially heterogeneous fields of Punjab and the fine-scale extractive dynamics of selective logging and smallholder land uses in Peru when compared to the existing 30 m resolution maps used for stratification. At the same time, the 30 m maps are useful for improving the precision of the sample-based estimates derived from high-resolution data and field visits by incorporating the wall-to-wall information correlated with the target variable in a regression estimate, which we demonstrate in the current study.

The monitoring of forest degradation is an area in which the resolution of the satellite imagery can play a significant role. Forest degradation is essentially deforestation occurring at a scale that has been historically difficult to map. Planet imagery affords us the ability to monitor land-cover change events that are by definition temporary and often not visible in Landsat data due to both the spatial and temporal limitations of the Landsat system. There were a number of examples of selective forest harvest in Peru that occurred and then 'disappeared' within a matter of months due to the closure of the surrounding canopy. At the spatial resolution of Landsat, we can often observe the access roads to selective forest harvest sites, but not the harvest degradation itself. In our example Figure 7, it is evident that it is possible to map tree removal at a scale undetectable at coarser resolutions that disturbs the canopy for a short period of time.

Some of the limitations of using PlanetScope data relate to the automated bulk search and data download features of the Planet catalog. For example, there are currently no standard tools for selecting a cloud-free image for each sample block with a pre-defined temporal frequency. The other limitation is data cost, although programs like NASA's Commercial Smallsat Data Acquisition program, which facilitated the current study, and Planet's Education and Research program, provide imagery for university-affiliated researchers, making these data accessible for research and education purposes.

The production of wall-to wall maps can be costly in terms of time and resources as the cadence and resolution of earth observation imagery increases. It is often not practical to create wall-to-wall maps using sub-10 m satellite imagery. Additionally, wall-to-wall maps have no statistical properties ensuring estimates without bias and with known uncertainty. By coupling Planet imagery with a targeted sampling strategy, the presented method allows us to produce land-cover estimates at the national, continental, and perhaps global scales based on mapping conducted over very high-resolution imagery in accordance with IPCC guidelines [2]. The overall result is an efficient area-estimation methodology for large area analyses.

**Author Contributions:** J.P. acquired the imagery, led the mapping in Peru, and wrote the initial manuscript. A.T. led the area calculations and statistical analysis and edited the manuscript. A.K. performed fieldwork and mapping for the Pakistan work. P.P. created the seasonal Landsat metrics and provided support in data management and analysis along with leading the algorithm development for the research. B.A. provided technical support in remote sensing analysis. M.C.H. conceptualized and supervised the work, providing technical assistance where necessary. A.L. supported image classification. All authors have read and agreed to the published version of the manuscript.

**Funding:** The authors wish to acknowledge NASA's Commercial Smallsat Data Acquisition program, NASA's Land-Cover and Land-Use Change program grant 20-LCLUC2020-0029, and the US government interagency SilvaCarbon program for providing support for this research.

**Data Availability Statement:** Derived block map data available on request, Planet imagery unavailable due to licensing restrictions.

**Conflicts of Interest:** The authors declare no conflict of interest.

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
