# Peer review of "Using Multi-Resolution Satellite Data to Quantify Land Dynamics: Applications of PlanetScope Imagery for Cropland and Tree-Cover Loss Area Estimation"

_remotesensing, doi:10.3390/rs13112191_

Round 1
Reviewer 1 Report
Authors have presented the application of Planetscope Imagery for sample-based area estimation of wheat in Pakistan and tree cover loss in Peru. However, I believe this paper is for a technical report not a scientific article. Therefore, I suggest the rejection of this paper.
* whilst authors have presented the literature on the use of remote sensing, they does not indicate what is the research gap, what is the research aim and objectives. Therefore, this paper does not necessarily address a scientific question.
* The results just presented the mapping of wheat and tree cover, but I cannot catch what research problem authors can address. Do authors want to the satisfaction of the requirements?
Author Response
We thank Reviewer 1 kindly for their contribution. Please see our response attached.

Reviewer 2 Report
This paper needs a lot of improvement if the authors wish to publish it. First of all the contents have to be changed with a specific focus on certain themes but not roaming around the definitional terms.

Author Response
We thank Reviewer 2 kindly for their contribution. Please see our response attached.

Reviewer 3 Report
This study used 2205 PlanetScope images to estimate wheat in Pakistan (Dec/2018 – Apr/2019) and 730 scenes (2017/2018) to estimate deforestation in Peru, based on supervised bagged classification trees. Below, my suggestions to improve this version of the manuscript:
- Most of the citations (12 out of a total of 19) belong to the same group of authors from the Department of Geographical Sciences of the University of Maryland. Authors should provide results of studies of land cover changes from other groups, at least in the contextualization of the work in the Introduction section.
- Fully-automated image processing is one of the major problems of the PlanetScope microsatellite images because of the poor radiometric quality control among the images from the same study area. It seems that the supervised bagged classification trees, considered by the authors in this paper, is a good strategy to overcome such difficulty. This point can be highlighted in the text since it is a strong point of the manuscript.
- Figures 1 and 2 can be improved by adding inset maps with locations in their respective countries. Overlaying the main maps in the satellite images, for example, RGB color composites of Landsat 8 images will be nice as well. In Figure 3, please add the total number of scenes per study area.
- Conclusion section is missing in the manuscript.
Author Response
We thank Reviewer 3 kindly for their contribution. Please see our response attached.

Round 2
Reviewer 3 Report
All my concerns from the previous version of the manuscript were well-addressed by the authors. I rate the new version af ready to be published.
Author Response
We thank Reviewer 3 very much for their time and effort in assessing our manuscript.